# Sensory Profile of Monferace Wine: An 'Old-Style' Vinification Approach for Grignolino, a Red Indigenous Italian Variety

Maria Carla Cravero [1,*], Federica Bonello [1], Andriani Asproudi [1], Silvia Gianotti [2], Mario Ronco [2] and Maurizio Petrozziello [1]

1   CREA, Council for Agricultural Research and Economics, Research Centre for Viticulture and Enology, Via P. Micca 35, 14100 Asti, Italy; federica.bonello@crea.gov.it (F.B.); andriani.asproudi@crea.gov.it (A.A.); maurizio.petrozziello@crea.gov.it (M.P.)
2   Associazione Monferace, Castello di Ponzano Monferrato, Piazza Vittorio Veneto 1, 15022 Ponzano Monferrato, Italy; info@monferace.it (S.G.); ronco@atlink.it (M.R.)
*   Correspondence: mariacarla.cravero@crea.gov.it

**Abstract:** Grignolino is an autochthonous Italian red variety cultivated in Piedmont (north-west Italy), used in high percentages (90–100%) to produce three main different DOC wines, generally consumed young. The Monferace project was born of an idea of 12 winegrowers wanting to create a new "old style" Grignolino red wine and inspired by ancient winemaking techniques of this variety. Monferace wine is produced following a discipline which gives some guidelines but no indications on the vinification technique or on the variety of wood to be used. The percentage of Grignolino grapes should be 100% and the ageing of 40 months, 24 of which are spent in wooden barrels of differing volumes. The aim of this work is the definition of the sensory profile of Monferace wines during ageing. The sensory analysis on 10 Monferace wines (2019 vintage) was assessed after approximately 11 months of ageing in wood. A trained panel carried out the wine sensory descriptive analysis (sensory profile) with a methodology derived from ISO norms. The results showed that all the wines were characterized by 16 attributes: color (garnet red, orange highlights), odor (rose, violet, nutmeg, pepper, blackberries, cherries, jam/marmalade, dry herbaceous, boisé-oak wood) and taste (acidity, bitterness, astringency, structure (body) and taste–olfactory persistence). Some attributes were, quantitatively, not statistically different: acidity, bitterness, astringency. All the other attributes discriminated the wines with different intensities, and each wine had a specificity. These preliminary results demonstrated the cohesion of sensory attributes among the wines, with individual distinctions within each product, and indicated that Monferace is a very promising wine style for the Grignolino variety.

**Keywords:** sensory analysis; Grignolino; wood ageing; Monferace; wine profile

## 1. Introduction

The Monferace project was born of an idea of 12 winegrowers wanting to create a new "old style" Grignolino red wine and inspired by an ancient winemaking technique of this autochthonous Italian variety, cultivated in Piedmont (north-west Italy). Monferace wine producers aim to enhance the best characteristics of the Grignolino wine, well known in the past [1]. As reported by [2], in 1946, Garino-Canina wrote that Grignolino wines, obtained from old vines and suitably aged, were comparable to the best Burgundy wines.

The Monferace Grignolino is cultivated in the geographical area identified in the Aleramic Monferrato, defined by the Po and Tanaro rivers, in the heart of Piedmont. The Monferace wine is characterized by a high tannin content, marked when young, which evolves over the years [3,4].

Monferace wine is produced according to certain guidelines [3], the most important being that the wine should be produced with 100% Grignolino grapes and can only be put on the market after a minimum of 40 months ageing—calculated from 1 November

of the vintage year—with at least 24 months spent in wooden barrels. The registered vineyards must be planted on limestone-silt-clay soils, of varying compositions, also with the natural presence of sandy sediments. The vineyards must be exclusively hilly, with suitable exposure to ensure optimal ripening of the grapes. The number of vines per hectare cannot be less than 4000. The culture methods are those traditional within the area (counter-espalier with upright vegetation). Pruning systems can be traditional guyot and low spurred cordon. The maximum yield of grapes must not exceed 7 tons per hectare [3]. In the guidelines, there are no specifications either on the vinification technique or on the type of wood to be used; it can be a barrique or a large wooden barrel of any origin (America or Europe), at the discretion of the producer. The average annual production of Monferace is approximately 20,000 bottles (data from the Monferace Association).

Grignolino is an ancient variety, also called Barbesino, probably cultivated in the Monferrato hills, an area which has produced Monferace since the XIII century [1].

In a study on the color and anthocyanin evaluation of eighteen red winegrapes [5], Grignolino was placed in the same group together with the most famous Nebbiolo, due to their low anthocyanin content (353 mg/L), and peonidin being their most prominent anthocyanin (58%), followed by malvidin (20%). Their grapes were found to be similar, with regard to the polymeric proanthocyanidins in skins and seeds [6], and to the total flavonoid and flavan-3-ols content of their seeds [7], though their flavanol profiles and contents differ [8]. According to a recent DNA-based genealogy reconstruction [9], Nebbiolo is the grandfather of Grignolino. Nebbiolo is a variety of prestigious aged wines, such as Barolo and Barbaresco [9]. On the other hand, Grignolino wines are generally not aged, but consumed young, the year after production [4]. There are three different DOC wines [10]. Grignolino d'Asti DOC is produced in the province of Asti (10,675 hL, data 2021 Valoritalia [11]), with grapes of the homonymous variety (90–100%) and Freisa (maximum 10%) variety. Grignolino del Monferrato Casalese is produced in the province of Alessandria (5037 hL, data 2021 Valoritalia [12]). This DOC is obtained with grapes of the homonymous variety (95–100%), and Freisa or Barbera (maximum 5%) varieties. Piemonte Grignolino DOC is produced with grapes of the homonymous variety (85–100%), as well as other varieties (<15%) cultivated in province of Asti, Alessandria and Cuneo. Grignolino can be incorporated in other DOC wines of Piedmont in a small percentage, alone or with other varieties (5–25%): Gabiano, Barbera del Monferrato Superiore, Barbera del Monferrato, Rubino di Cantavenna and Colli Tortonesi rosso [10].

This work was carried out as part of the SESAMO project "Studio delle peculiarità Enologiche, Storiche, Ambientali e viticole del Monferrato 'Aleramico' per la valorizzazione del Grignolino affinato in legno", (RF = 2019.2337, founded by the Cassa di Risparmio di Torino Foundation, Turin, Italy), a study of the chemical and sensorial peculiarities of the Grignolino Monferace wines. In this study the definition of the sensory profiles of a selection of commercially 2019 Monferace wines, approximately after one third of the required ageing, is illustrated. Some data were partly anticipated in a previous study [13]. Moreover, these results are compared to the sensory profiles of young Grignolino wines described in previous studies, [14,15] and of some Monferace wines [16].

## 2. Materials and Methods

A total of 10 Monferace wines (2019 vintage) were evaluated after approximately 11 months of ageing in wood, about one third of the required ageing. All the wines—except wine 9, which was produced with a short maceration—were produced with a very long submerged cap maceration (2–3 months) and a spontaneous fermentation. At the end of the fermentation, the wines 1, 2, 3, 4, 7, 8 and 10, were put in a new 5 hL cask (tonneau); wine 5 in a used 15 hL barrel; wine 6 in a new 10 hL barrel and wine 9 in a used 5 hL cask (tonneau).

The wine sensory descriptive analysis (sensory profile) was performed by a trained panel (5 males and 8 females), following the procedure described in previous papers [17,18] and derived by the ISO norms.

The wines were evaluated using ISO (3591-1977)-approved glasses in an ISO (8589-2007) tasting room. In each sensory session, 5 wines were served (50 mL) blindly, in a randomized order and identified with a three-digit code.

All the wines were tasted in a preliminary tasting session, to define the odor descriptors with the help of a predefined odor list [19], with the three levels of specificity, from the most generic 1st level (i.e., Fruity), medium generic 2nd level odor (i.e., Berries), to the most specifically termed 3rd level (i.e., raspberries, blackberries, strawberries, blueberries, red currants). The choice of descriptors was made on the identification frequencies. The color attributes and the 2nd level odor descriptors were chosen if their frequency of identification was higher than (n° assessors × n° wines/2), and the 3rd level descriptors if their frequency was higher than (n° assessors × n° wines/4). The taste and mouthfeel attributes evaluated were acidity, bitterness, astringency, structure (body) and taste–olfactory persistence.

The chosen attributes were confirmed by presenting the panel with appropriate standards and measured twice in double using unstructured scales (0–100) in two different tasting sessions.

Qualitative and quantitative sensory data were collected using the FIZZ software (Biosystèmes, Couternon, France).

The quantitative sensory results (sensory profiles) were processed with ANOVA and Tukey's test ($p = 95\%$), considering the factors wine, assessor and sensory session and their interactions, and with PCA using XLSTAT® software, version Sensory, 2020, 2.2 (Addinsoft, New York, NY, USA).

## 3. Results

### 3.1. Descriptors

The color attributes and the 2nd level odor descriptors were chosen if their frequency of identification was higher than 60 (12 assessors × 10 wines/2), and the 3rd level descriptors if their frequency was higher than 30 (12 assessors × 10 wines/4). Only 12 of the 13 assessors participated in all sessions of attribute identification. The panel identified 16 attributes in the 10 wines: 2 for the color (garnet red, orange highlights), 7 3rd level odor descriptors (rose, violet, nutmeg, pepper, blackberries, cherries, jam/marmalade), 2 2nd level odor descriptors (dry-herbaceous and boisé-oak wood) and 5 taste and tactile sensation attributes (acidity, bitterness, astringency, structure (body) and taste–olfactory persistence).

The panel (13 assessors) identified one more specific odor attribute in only wine 2 (vanilla) and wine 7 (smoked-roasting), whose frequencies were, respectively, 8 (8/12 assessors) and 7 (7/12 assessors).

### 3.2. Sensory Profiles

The sensory profiles of the 10 wines were all defined by the panel (Figure 1).

The ANOVA results are shown in Table 1.

The factor wine resulted statistically significative (ANOVA and Tukey test, $p = 95\%$) for all the attributes except acidity, bitterness and astringency.

The session and the interaction between wine and session were never statistically significant.

Some descriptors were not so robust because some interactions were significative: assessor × session and assessor × wine for garnet red and pepper, assessor × wine for cherries and jam-marmalade, assessor × wine for dry herbaceous and assessor × session for astringency.

Many attributes differentiated the wines with different intensities, from two groups in the case of rose, nutmeg and dry-herbaceous, to six groups for boisé-oak wood (Table 2).

**Figure 1.** The sensory profile of the 10 Monferace wines.

**Table 1.** ANOVA results of the sensory quantitative attributes.

| | Assessor | Wine | Session | Assessor × Session | Assessor × Wine | Wine × Session |
|---|---|---|---|---|---|---|
| Garnet red | *** | *** | n.s. | *** | *** | n.s. |
| Orange highlights | *** | *** | n.s. | n.s. | n.s. | n.s. |
| Rose | *** | ** | n.s. | n.s. | n.s. | n.s. |
| Violet | *** | *** | n.s. | n.s. | n.s. | n.s. |
| Nutmeg | *** | *** | n.s. | n.s. | n.s. | n.s. |
| Pepper | *** | *** | n.s. | *** | ** | n.s. |
| Blackberries | *** | *** | n.s. | n.s. | n.s. | n.s. |
| Cherries | *** | *** | n.s. | n.s. | ** | n.s. |
| Jam-marmelade | *** | *** | n.s. | n.s. | *** | n.s. |
| Boisé-oak wood | *** | *** | n.s. | n.s. | n.s. | n.s. |
| Dry-herbaceous | *** | ** | n.s. | n.s. | *** | n.s. |
| Acidity | *** | n.s. | n.s. | n.s. | n.s. | n.s. |
| Bitterness | *** | n.s. | n.s. | n.s. | n.s. | n.s. |
| Astringency | *** | n.s. | n.s. | *** | n.s. | n.s. |
| Structure | *** | *** | n.s. | n.s. | n.s. | n.s. |
| Taste–olfactory persistence | *** | *** | n.s. | n.s. | n.s. | n.s. |

**, *** significant at $p < 0.01$ and $p < 0.001$, respectively; n.s. means not significant.

**Table 2.** Results of the Tukey test of the sensory quantitative attributes with significant difference at ANOVA.

| Wine | Garnet Red | *** | Wine | Orange Highlights | *** | Wine | Rose | ** | Wine | Violet | *** | Wine | Nutmeg | *** |
|---|---|---|---|---|---|---|---|---|---|---|---|---|---|---|
| 2 | 60.1 | A | 2 | 67.4 | A | 5 | 45.6 | A | 5 | 45.7 | A | 7 | 41.9 | A |
| 6 | 54.7 | AB | 9 | 60.8 | AB | 10 | 38.9 | AB | 10 | 45.0 | A | 6 | 38.8 | A |
| 7 | 54.5 | AB | 7 | 57.9 | ABC | 3 | 37.6 | AB | 3 | 41.6 | AB | 10 | 36.0 | A |
| 1 | 53.1 | ABC | 8 | 55.3 | ABC | 7 | 37.5 | AB | 6 | 35.4 | AB | 5 | 35.3 | AB |
| 8 | 51.4 | ABC | 6 | 53.7 | BC | 1 | 37.0 | AB | 7 | 34.6 | ABC | 2 | 33.0 | AB |
| 4 | 50.0 | ABCD | 1 | 49.2 | BCD | 2 | 31.8 | AB | 1 | 30.4 | BCD | 3 | 31.3 | AB |
| 5 | 46.0 | BCD | 4 | 46.7 | CD | 6 | 31.8 | AB | 2 | 28.2 | BCD | 1 | 28.1 | AB |
| 10 | 41.4 | CD | 5 | 38.7 | D | 9 | 28.2 | B | 4 | 20.4 | CD | 4 | 27.6 | AB |
| 9 | 38.3 | D | 3 | 20.4 | E | 4 | 28.1 | B | 9 | 19.9 | D | 9 | 20.7 | B |
| 3 | 37.5 | D | 10 | 17.5 | E | 8 | 26.1 | B | 8 | 19.4 | D | 8 | 20.5 | B |

**Table 2.** *Cont.*

| Wine | Pepper | *** | Wine | Black-berries | *** | Wine | Cherries | *** | Wine | Jam-Marme-lade | *** |
|------|--------|-----|------|---------------|-----|------|----------|-----|------|----------------|-----|
| 3 | 43.8 | A | 10 | 56.7 | A | 10 | 51.1 | A | 10 | 52.1 | A |
| 7 | 43.2 | AB | 5 | 43.3 | B | 5 | 49.4 | AB | 5 | 44.9 | AB |
| 6 | 39.6 | AB | 1 | 40.4 | B | 7 | 43.1 | AB | 6 | 44.8 | AB |
| 5 | 38.4 | AB | 6 | 38.2 | B | 6 | 42.2 | AB | 7 | 39.5 | BC |
| 10 | 37.7 | AB | 7 | 36.5 | B | 1 | 40.4 | ABC | 2 | 38.9 | BC |
| 1 | 31.3 | ABC | 2 | 35.2 | B | 2 | 38.1 | BC | 1 | 35.4 | BCD |
| 2 | 31.1 | ABC | 3 | 26.3 | C | 3 | 28.7 | CD | 4 | 28.7 | CDE |
| 4 | 30.0 | BC | 4 | 23.7 | CD | 4 | 24.5 | D | 3 | 26.0 | DE |
| 9 | 23.6 | C | 8 | 18.3 | CD | 9 | 20.4 | D | 8 | 19.7 | E |
| 8 | 20.0 | C | 9 | 16.7 | D | 8 | 185 | D | 9 | 19.3 | E |

| Wine | Dry-Herbace-ous | ** | Wine | Boisé-Oak Wood | *** | Wine | Structure | *** | Wine | Taste–Olfactory Persis-tence | *** |
|------|-----------------|-----|------|----------------|-----|------|-----------|-----|------|------------------------------|-----|
| 6 | 42.2 | A | 2 | 64.1 | A | 10 | 59.3 | A | 10 | 62.5 | A |
| 3 | 38.4 | A | 6 | 58.8 | A | 5 | 55.8 | AB | 5 | 60.2 | A |
| 7 | 38.0 | A | 7 | 55.8 | AB | 6 | 52.8 | AB | 3 | 56.8 | AB |
| 10 | 33.7 | AB | 10 | 43.0 | BC | 3 | 52.3 | AB | 6 | 55.3 | AB |
| 2 | 32.2 | AB | 3 | 40.0 | CD | 2 | 52.2 | AB | 2 | 54.1 | AB |
| 4 | 30.7 | AB | 5 | 33.3 | CDE | 7 | 51.8 | AB | 7 | 53.7 | AB |
| 5 | 30.5 | AB | 4 | 27.9 | DEF | 4 | 49.7 | AB | 1 | 53.4 | AB |
| 9 | 30.4 | AB | 1 | 25.5 | DEF | 1 | 49.3 | AB | 8 | 47.6 | BC |
| 1 | 30.0 | AB | 8 | 19.5 | EF | 8 | 45.3 | BC | 4 | 46.6 | BC |
| 8 | 23.5 | B | 9 | 15.9 | F | 9 | 38.1 | C | 9 | 39.4 | BC |

**, *** significant at $p < 0.01$ and $p < 0.001$, respectively. Different letters indicate significant statistical differences with ANOVA and Tukey's test ($p = 95\%$).

Almost each wine had a specificity: wine 5 had the highest intensity for rose, wine 10 for fruity attributes (blackberries, cherries), wine 2 for oak wood together with vanilla (Figure 2), wine 6 for dry-herbaceous, wine 7 for smoked-toasting (Figure 3) and wine 3 for pepper.

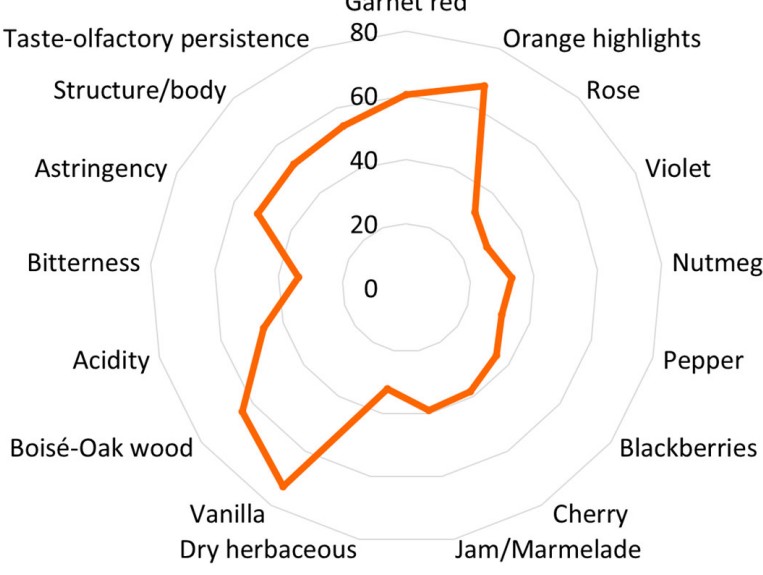

**Figure 2.** The average sensory profile of wine 2.

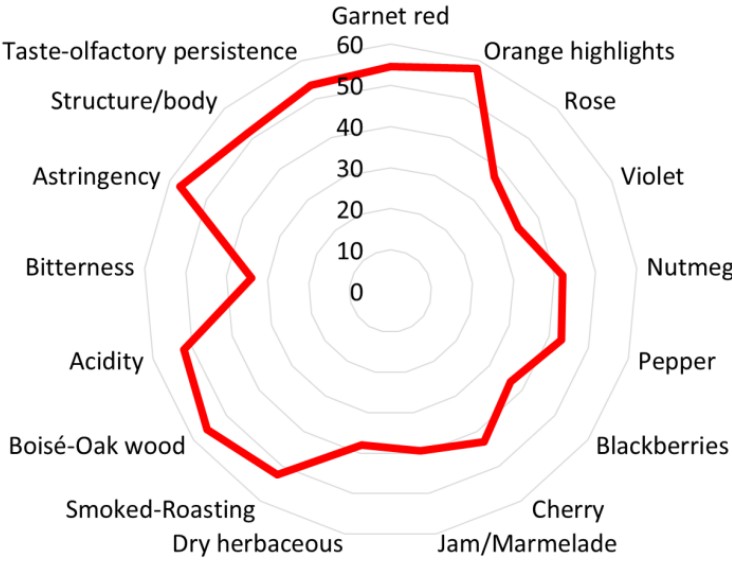

**Figure 3.** The average sensory profile of wine 7.

Wines 8 and 9 had the lowest intensities for many attributes, and the profile of wine 1 was very similar to the average profile of all 10 wines (Figure 4).

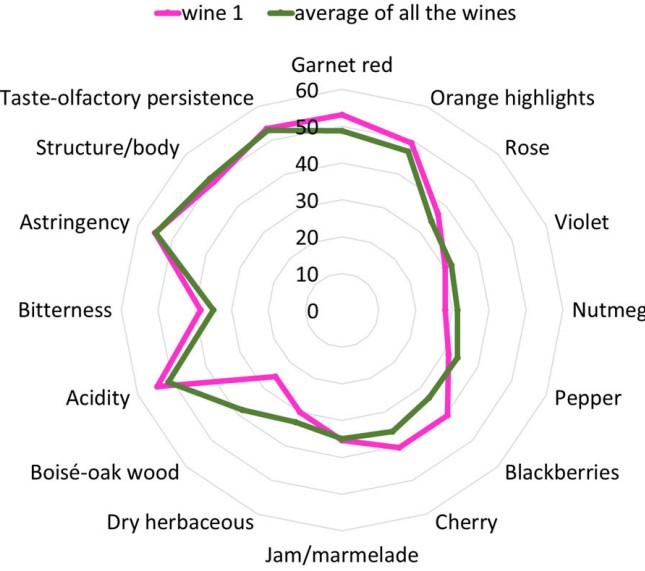

**Figure 4.** The average sensory profile of the 10 wines and the sensory profile of wine 1.

In Figure 5 the PCA shows the distribution of the wines.

The first component explains 53% of the variance and the second component explains 17%. The samples, 4, 8 and 9, are separated from the others, with more intense orange highlights and an olfactory profile characterized by low intensities of many attributes (Figure 1 and Table 1).

Wines 3, 5 and 10 show the highest intensities for rose and violet odors, whereas wines 6 and 7, for dry-herbaceous and boisé-oak wood (Figure 1 and Table 1). The first component can explain the differing complexities of the wines, whereas the second component explains the evolution from fresh wines with floral attributes (rose, violet) towards wines where the ageing in wood is more evident.

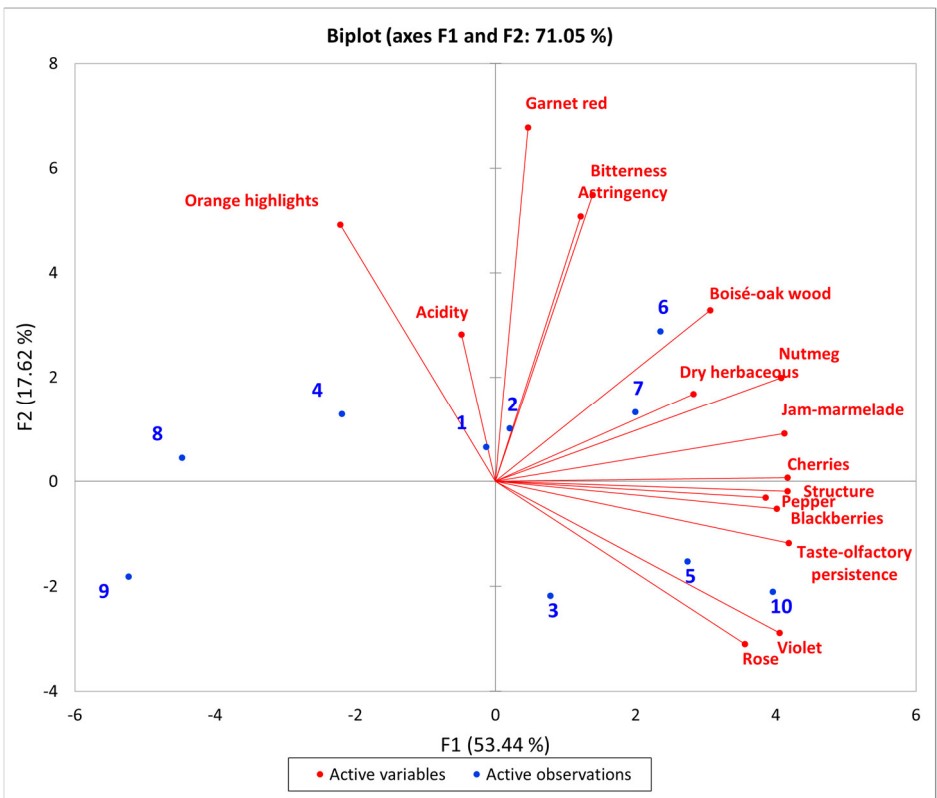

**Figure 5.** PCA results.

## 4. Discussion

A previous study [14] was carried out on a suitable number of commercial Grignolino wines, of the three main DOC wines (16 wines of 2000 and 20 of 2001) evaluated the year after the vintage. In those samples, the color was garnet red with orange highlights, and the olfactory descriptors were violet-rose, geranium flower, pepper, raspberry and straw-hay. The taste attributes were acidity, bitterness, astringency, structure and taste–olfactory persistence. The quantitative evaluations showed a few significative quantitative differences: for the color attributes and raspberry odor in 2000 and 2001 wines, rose-violet and taste–olfactory persistence for the 2000 wines and astringency for the 2001 vintage only. Most likely, these quantitative differences were due not only to the vintage, but also to the differing winemaking techniques of the producers.

All those taste descriptors, and some of those olfactory attributes (rose, geranium flower, pepper, raspberry), were confirmed in non-aged Grignolino wines produced in purity in a recent evaluation [15].

These preliminary results show that Monferace wines, after 11 months of ageing, have some descriptors which also characterize the non-aged Grignolino wines; these are garnet red and orange highlights for the colour, rose, violet, pepper, raspberry and dry-herbaceous (straw-hay) for the odors and the same descriptors for the taste. Monferace wines exhibit a more complex sensory profile. In addition to the wood descriptors, some other odor attributes, such as nutmeg, cherry, jam/marmelade and blackberry instead of raspberry, were identified. The olfactory attributes are in line with those of 2012 (4 years of ageing) and 2015 (2 years of ageing) Monferace wines, characterized by wood, boisée, floral, cherry, wild berries, caramel and spices [16].

The spicy notes of Grignolino, and especially the pepper attribute, could be established as markers for Grignolino wines, since unpublished preliminary data highlight that they could be related to the presence of rotundone, a sesquiterpene known for its extremely low perception threshold and an intense peppery note [20].

The Monferace wines show differences as regards the color, with different intensities for garnet red and orange highlights, the odor, the structure, and the taste-olfactory persistence, showing a different evolution after the same duration of ageing in wood. Some products retain high intensities of floral notes (rose and violet) after 11 months of ageing, whereas in other wines the effect of wood is more evident.

The sensory results can be influenced by varying conditions of wine ageing, such as whether used or new wood containers are utilized, and the volumes of these containers (5 hL or 15 hL or 10 hL). Each producer can determine the technique at their own discretion, since there are no specifications on winemaking and ageing procedures in the guidelines on Monferace, but only on ageing time. For example, in the PCA, wine 9 was separated from the other wines (Figure 5) and exhibited the lowest intensities for almost all attributes (Figure 1); it was aged in a used 5 hL cask, and it is the only wine produced with a short fermentation.

The ageing in wood enables the enhancement of the Grignolino variety, and the attainment of a product which can be appreciated by a distinct category of consumers.

Moreover, the 10 wines showed both a uniformity in the 16 sensory attributes, as well as individually distinguishing features within each product, indicating that Monferace is a very promising wine style for the Grignolino variety.

**Author Contributions:** Conceptualization: M.P.; methodology M.C.C. and F.B.; formal analysis, M.C.C. and F.B.; Writing—original draft preparation M.C.C.; review and editing A.A., F.B., M.P. and M.C.C.; supervision, M.P., M.R. and S.G.; Project administration: M.P.; Funding acquisition: M.P., M.R. and S.G. All authors have read and agreed to the published version of the manuscript.

**Funding:** The SESAMO project "Studio delle peculiarità Enologiche, Storiche, Ambientali e viticole del Monferrato 'Aleramico' per la valorizzazione del Grignolino affinato in legno", (RF = 2019.2337) has been founded by the Cassa di Risparmio di Torino Foundation (Turin, Italy).

**Data Availability Statement:** Not applicable.

**Acknowledgments:** Acknowledgements to the Monferace Association and to the assessors of the panel of CREA, Asti. All individuals included in this section have consented to the acknowledgement.

**Conflicts of Interest:** The authors declare no conflict of interest.

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
