# Peer review of "Sensory Profile of Monferace Wine: An ‘Old-Style’ Vinification Approach for Grignolino, a Red Indigenous Italian Variety"

_beverages, doi:10.3390/beverages9020046_

Round 1

Reviewer 1 Report

The manuscript “Monferace a New “Old Style” for Grignolino Wine, an Autochthonous Italian Variety: Unity in Diversity” presents an innovative proposal by carrying out a complete sensory evaluation of a new variety of wine inspired by ancient production techniques.

Below are the considerations:

- In the introduction there are few bibliographic citations. Mention how is the market and the production of the wine produced by this grape.

- In the last paragraph of the introduction mentions that the manuscript aims to study the chemical and sensory peculiarities. What would be the chemical characteristics studied.

- Materials and methods: Was the study approved by an Ethics Committee? To clarify.

Is the number of evaluators statistically significant?

Briefly describe the descriptors used by the evaluators for each attribute.

- Discussion: Could some of these attributes be defined as markers for this style of wine? Discuss the possibility of assigning designation of origin.

- References are few in number and more than half have more than ten years of publication.

Reviewer 2 Report

The research work presented in this manuscript is very poor. The authors should reconsider the design of their experiments for future work.

-        The authors should rewrite the title of the manuscript to clarify the readers which is the variety of autochthonous grapevine.

-        The abstract needs revision. The methodology used is not included.

-        The introduction section is very poor. They must expand it and include the corresponding citations of bibliographical references. The authors include information that is not referenced.

-        The objective proposed by the authors does not correspond to what the authors present. They have not presented results of the chemical composition of the wines.

-        In the material and methods section, the description of the wine production is missing.

-        The tasting sheets used should be provided.

-        The number of tasters is low.

-        There would be a lack of discussion of the results with other previous Works.

Reviewer 3 Report

The communication " Monferace a New “Old Style” for Grignolino Wine, an Autochthonous Italian Variety: Unity in Diversity " submitted to the Beverages journal summarises the sensory analysis of 10 Grignolino Monferace wines. The production of excellent Monferace wines requires good quality Grignolino grapes and ageing in wood under the right conditions. The participants in the Monferace project have carried out sensory profiles of 10 wines to get an overview of the sensory quality parameters of these wines.

My comments on the manuscript:

Until the reader reaches the Discussion, it is not clear whether the work done serves as more than a data description. Why are sensory judgements published after 11 months of maturation if at least 40 months are required? Then, in the Discussion, the analyses of the 2000-2001 wines are described.

·         I would ask the authors to highlight in the introduction the thoughts between lines 149-159 of the discussion. The objectives should be clarified and completed in lines 62-65. If the reader knows from the outset that the authors are going to compare their results with something, the reader will be less critical and will have the impression that it is only a report of data.

·         Some people may not be well informed about Italian grape varieties. Please indicate in the abstract and in the introduction that Grignolino is a red wine grape variety.

·         Could the organoleptic differences in the 2000-2001 wines be due to the different winemaking technology and vintage? At least mention in the manuscript the authors' opinion on this.

·         Part of the conclusion is that barrel ageing can have an impact on the development and organoleptic quality of wines. It would also be interesting for the reader to have information on the type of barrels used for ageing. New or previously used oak barrels, low or high capacity? Fully or partially filled barrels with wine. For winemakers, these ageing parameters can be useful information.

The article is correctly written. The data and analyses are well presented. The tables and graphs are easy to follow and the way the data are evaluated is relevant.

The methodology is sufficiently detailed and replicable.

It would be more useful to carry out sensory and analytical tests on the wines mentioned in the manuscript at the end of the project, as they are not likely to be of widespread scientific interest in this form.

Reviewer 4 Report

Although the manuscript deals with a very specific and local topic, it could be worth publishing. However, the results are very preliminary and the discussion is poor, so authors should deeply improve this section. In addition, the number of references is very low and almost all of them are local. I suggest to include more international references that support the results.

I have some other aspects to be improved in the manuscript:

- The title is not adequate because it does not reflect the content of the manuscript. 

- Line 62. Authors should include some more information about this SESAMO project if they want to mention it (funding institution, name, etc.)

- Lines 76-78. What are second and third levels descriptors? Please, explain this fact. 

- Line 98. It should be "and".

- Line 99. What does authors mean with Boisé-oak? Why not simply oak? Please, explain.

- Line 139. The percentage is not 51%.

Round 2

Reviewer 2 Report

-        The authors should rewrite the title of the manuscript. The The title of your manuscript should be concise, specific and relevant.  

-        The abstract should be rewritten following the manuscript preparation guidelines published by the journal: “The abstract should be a single paragraph and should follow the style of structured abstracts, but without headings: 1) Background: Place the question addressed in a broad context and highlight the purpose of the study; 2) Methods: Describe briefly the main methods or treatments applied. Include any relevant preregistration numbers, and species and strains of any animals used. 3) Results: Summarize the article's main findings; and 4) Conclusion: Indicate the main conclusions or interpretations”.

-        The introduction is not comprehensible to scientists working outside the topic of the paper. The information is not ordered. The objectives of the manuscript are not clearly written. They include information that needs to be contrasted with bibliographic citations (lines 35-38; lines 39-44; line 50; lines 60-63; lines 66-72).

-        The authors should describe in more detail the elaboration of the wines, including details such as the type of wood used for aging.

-        This work includes experimental design errors, aging comparisons are not performed in duplicate or triplicate.

Author Response

Please, see the attached file

Reviewer 4 Report

Authors have slightly improved the manuscript, but it is true that it is a short communication, so in that case it could be acceptable for publication.

However, I still think that the title suggested by authors is not adequate because it is not illustrative of the contents of the article. So I would suggest to replace it by a different one.

Author Response

Please, see the attached file
